# Redox Chemistry of the Subphases of α-CsPbI_2_Br and β-CsPbI_2_Br: Theory Reveals New Potential for Photostability

**DOI:** 10.3390/nano13020276

**Published:** 2023-01-09

**Authors:** Lavrenty Gennady Gutsev, Sean Nations, Bala Ramu Ramachandran, Gennady Lavrenty Gutsev, Shengnian Wang, Sergei Aldoshin, Yuhua Duan

**Affiliations:** 1Institute for Micromanufacturing, Louisiana Tech University, Ruston, LA 71272, USA; 2Federal Research Center for Problems of Chemical Physics and Medicinal Chemistry of RAS, Semenov Prospect 1, Chernogolovka 142432, Russia; 3Department of Physics, Florida A&M University, Tallahassee, FL 32307, USA; 4National Energy Technology Laboratory, United States Department of Energy, Pittsburgh, PA 15236, USA

**Keywords:** density functional theory, inorganic perovskite solar cells, mixed perovskite solar cells, photovoltaics, redox chemistry, nanoinclusions

## Abstract

The logic in the design of a halide-mixed APb(I_1−x_Br_x_)_3_ perovskite is quite straightforward: to combine the superior photovoltaic qualities of iodine-based perovskites with the increased stability of bromine-based perovskites. However, even small amounts of Br doped into the iodine-based materials leads to some instability. In the present report, using first-principles computations, we analyzed a wide variety of α-CsPbI_2_Br and β-CsPbI_2_Br phases, compared their mixing enthalpies, explored their oxidative properties, and calculated their hole-coupled and hole-free charged Frenkel defect (CFD) formations by considering all possible channels of oxidation. Nanoinclusions of bromine-rich phases in α-CsPbI_2_Br were shown to destabilize the material by inducing lattice strain, making it more susceptible to oxidation. The uniformly mixed phase of α-CsPbI_2_Br was shown to be highly susceptible towards a phase transformation into β-CsPbI_2_Br when halide interstitial or halide vacancy defects were introduced into the lattice. The rotation of PbI_4_Br_2_ octahedra in α-CsPbI_2_Br allows it either to transform into a highly unstable apical β-CsPbI_2_Br, which may phase-segregate and is susceptible to CFD, or to phase-transform into equatorial β-CsPbI_2_Br, which is resilient against the deleterious effects of hole oxidation (energies of oxidation >0 eV) and demixing (energy of mixing <0 eV). Thus, the selective preparation of equatorial β-CsPbI_2_Br offers an opportunity to obtain a mixed perovskite material with enhanced photostability and an intermediate bandgap between its constituent perovskites.

## 1. Introduction

Lead-based perovskite solar cells are currently a very actively studied and rapidly evolving area of materials science research [1]. The scientific community’s interest in perovskites is largely due to their rapidly improving solar power conversion efficiencies (PCEs), which are now being reported to be as high as 25% [2]. In terms of PCE, this means that certified single-junction perovskite technology is comparable to the traditional state-of-the-art crystalline silicon photovoltaics (PV). Such an extraordinary performance is related to the low rate of radiative bimolecular recombination [3], long carrier lifetimes and large carrier lengths. It should be noted that the PCE directly correlates with the charge-carrier lifetime of the material. Unfortunately, the high PCEs of lead-based perovskites typically only last a few thousand hours [4,5,6], after which they degrade via a variety of mechanisms.

Paradoxically, perovskite materials are also highly defect-tolerant, particularly to the dominant neutral Schottky defects which are known to be plentiful in the bulk of the material [7]; however, these defects generally create shallow traps in the band gap while also being the source of ionic disorder. Defects which do act as traps appear to often not be as problematic as calculations would suggest; for example, the dangling bonds present on a undercoordinated lead after a Br vacancy is introduced into CsPbBr_3_ exhibit a dynamic, oscillating behavior at the picosecond timescale. This is indicative of the fact that phonon–electron coupling can detrap the carriers despite the presence of a deep trap [8]. The soft, flexible nature of the material is responsible for rapid structural fluctuations which allow the local band structure to dynamically change, which consequently greatly slows recombination and leads to an increased carrier diffusion length [9]. In the case of organic–inorganic perovskites (MAPbX_3_ and FAPbX_3_), this seems to be related to the counter cations, which, while not being directly responsible for band gap behavior, screen carriers from being trapped or recombined [10], and thus the theoretical analysis of these materials is consequently more challenging; notably, heterostructures, in particular, show promise in this regard [11].

It would appear that defects by themselves do not degrade the material’s PV properties, but rather it is the accumulation of deep traps at the surfaces, grains and interfaces that should be analyzed more closely. It should be noted that the perovskite PV stacks go through some degradation even when rationally engineered and well-encapsulated to prevent them from being degraded by environmental factors [12], which directly evidences the importance of preventing the photodegradation of the material. In the case of “pristine” materials based on a single halide, the tendency for the material to photodegrade has a few proposed explanations including the inherent volatility of the organic cations [13], surface-based degradation due to an accumulation of vacancy defects [14] and self-oxidation via charged Frenkel vacancy–halogen interstitial pairs generated by the coupling of carriers with the perovskite lattice [15]. 

An important observation pertaining to perovskites is the fact that the halide vacancy concentration increases dramatically upon illumination [15,16,17]. This leads to an increased ionic conductivity (σ_ion_) upon illumination. This observation has been proposed to be caused by an oxidative reaction involving the coupling of a hole carrier with an iodine site. The iodine moves into the interstitial site and leaves it as a charged vacancy:(1)$$I_{I}^{x}+h^{•}\to I_{i}^{x}+V_{I}^{•}$$

Such a localized charge-redistribution process is mediated by a small hole polaron and is likely the most common type of charged Frenkel defect (CFD). It is notable that hole polarons tend to have a significantly lower binding energy than the electron polarons and tend to be more localized in nature [10]. Theoretical work has suggested that self-trapping defects leading to localized halogen dimers (V centers, commonly called color centers [18]) may lead to the formation of highly mobile self-trapped excitons [19]. Theoretical studies suggest that shallow trapping is dominant in bulk while deep hole polarons are energetically expensive to generate [20]. The formation of I_2_^−^ at high interstitial [21] concentrations has been suggested to occur at high hole concentrations [19] by such a mechanism:(2)$$I_{I}^{x}+2h^{•}\to I_{i}^{•}+V_{I}^{•}$$

Charge loss at surfaces and interfaces seems to be particularly problematic for the stability of the material [22]. Particularly, redox chemistry at the surface can lead to the formation of more hole polarons, which can then penetrate back into the material to cause the formation of more Frenkel defects by mechanism (1) creating a self-propagating loop of photodegradation. We note that it is also possible for Frenkel defects to form in a charge-balanced way, which creates an $I_{i}^{\prime }/V_{I}^{•}$ pair.

In our recent paper [23], we described the effects of these deleterious CFDs arising from redox chemistry in the material migrating to the surface. In particular, the formation of the halide vacancy was associated with the creation of a metallic Pb-rich phase and the formation of the interstitial iodine was associated with the production of triiodide I_3_^−^, a superhalogen [24,25]. The mixed surface, having a higher work function and a lower iodine concentration, ought to have inhibited the formation of triiodide; however, halide-mixed perovskite materials offer an additional challenge to overcome: they phase-segregate upon illumination via the so-called “Hoke Effect” [26]. In the experimental part of the work [23], we found that the recombination of the Pb^0^ and I_3_^−^ defects at the interfaces of the material led to the dominance of the CsPbI_3_ phase at the grain boundary, thus negating the potential advantages of the mixed CsPbI_2_Br material. Our density functional theory (DFT) calculations, however, left some questions open, since we did not observe a noticeably large difference in the favorability of CFD formation for iodine-based defects versus bromine-based defects in α-CsPbI_2_Br, which led us to conclude that a more thorough analysis of the topic was necessary. It should be noted that a unified theory for light-induced segregation in mixed halides has recently been proposed [27] and supported by experimental evidence [28]. The theory indicates that light segregation ought to be less favorable for smaller cell values (Cs < MA, FA) and that CsPbI_2_Br should be stable in its mixed form since calculations indicate that $\mathrm{CsPb}{\left(I_{1-x}{\mathrm{Br}}_{x}\right)}_{3}$ ought to be stable up to x = 0.43 at room temperature and 1 sun of illumination. This supports our expectation that CFDs may be the primary cause of instability for CsPbI_2_Br.

The free energy of mixing of a mixed perovskite, while being an extremely important property, is not sufficient to fully describe the photostability, since there are materials, such as CsPbI_2_Br, which ought to be photostable, but which segregate anyways. This motivates our current report, where we study the connection between the electronic structure of the perovskite, the geometrical features, and the stability towards photodegradation for pure cubic CsPbI_3_, the halide-mixed black phase α-CsPbI_2_Br [29] and β-CsPbI_2_Br [30]. It should be noted that the cubic phase of CsPbI_3_, while experimentally unstable at room temperature in its bulk phase [31], may be stabilized in quantum dots [32] and also serves as a good reference point for comparison for CsPbI_2_Br as well as a relatively simple and robust model since we do not have to consider the orientation of the organic cation, which can greatly change the calculated defect formation energies in perovskite materials [33,34]. CsPbI_2_Br has been noted for its superior thermodynamic stability [35] and is also somewhat closer to the ideal Goldschmidt tolerance when compared to CsPbI_3_ [29] because of improved phase stability. For photovoltaic applications, α-CsPbI_2_Br and β-CsPbI_2_Br are prospective and recent experimental work has demonstrated the possibility of stabilizing α-CsPbI_2_Br via dilute FeCl_2_ incorporation [36]. Previous calculations have demonstrated that CsPbI_2_Br exhibits a *p*-type behavior due to the high formation energies of donor defects [29], and in the Br-poor phase, it exhibits a Fermi pinning of the Br interstitial defects to the halide vacancy. Meanwhile, surface-based calculations have shown that acceptor interstitial halide defects have very low formation energies, and the donor vacancies are generally quite unfavorable to form [37]. Presently, we focus on understanding the destructive effects of holes on the mixed inorganic perovskite material.

## 2. Materials and Methods

We performed DFT simulations using the Vienna Ab initio Simulation Package (VASP) [38]. We utilized the PBE generalized gradient approximation (GGA) [39], a D3 dispersion correction [40] and the projector augmented-wave (PAW) pseudopotentials [41,42]. All our calculations had an energy cutoff of 600 eV. This combination of functional and dispersion correction has previously been shown to provide very good agreement with the experimental lattice constants of CsPbI_3_ and RbPbI_3_ and also showed good agreement with the magnitude of the band gap without spin–orbit corrections due to error cancellation [43]. We also included nonspherical contributions to the gradient inside of the PAW spheres. This correction has recently been reported to change the energy by as much as 30 meV/f.u. for perovskite oxides [44].

The cubic CsPbI_3_ and CsPbI_2_Br unit cells were optimized and then used to build 3 × 3 × 3 supercells with the Atomic Simulation Environment (ASE) Python tools. In the case of CsPbI_3_, we also built a 4 × 4 × 4 supercell to investigate the effects of the supercell size. For the 3 × 3 × 3 supercells, we used k-grids of 3 × 3 × 3 and in the case of 4 × 4 × 4 supercells we used 2 × 2 × 2 sampling of the k-space. To ensure the sufficiency of this method, we tested the 3 × 3 × 3 supercell with 5 × 5 × 5 sampling of the k-space and found that the pristine-cell oxidation energy (Equation (3)) changed by less than 0.01 eV. In the unit cell cases, we used 6 × 6 × 6 k-grids or better. We allowed both the atomic positions and cell dimensions to relax without constraints, and also recalculated some of the hole-coupled oxidations with the MGGA functional HSE06 [45] to crosscheck our results. Previous calculations indicated that our PBE+D3 method was a reasonable compromise for calculating defect formation energies [46].

To compare the oxidative ability of each system we used the expression:(3)$$\Delta E_{ox}=\Delta E+q\left(E_{F}+E_{VBM}\right)+E_{corr}$$
(4)$$\Delta E=E\left(D^{q}\right)-E\left(D^{0}\right)$$
where $E\left(D^{0}\right)$ indicates the energy of the neutral slab and $E\left(D^{q}\right)$ indicates the energy of the charged slab. $E_{VBM}$ was extracted from band structure calculations at the kpoint corresponding to the direct band gap E_g_, the k-path of the band structure was determined with the help of AFLOW [47]. Finally, $E_{F}$ is the Fermi energy, which may vary from 0 to E_g_, and E_corr_ is the electrostatic correction. The simplest electrostatic correction is the Makov–Payne correction (the 1st order):(5)$$E^{q}=\frac{q^{2}\alpha }{2\varepsilon L}\;$$
where α is the Madelung constant, ε is the dielectric constant of the pristine bulk and L is the lattice constant. This relatively basic charge-corrective method has recently been demonstrated to show good convergence versus the supercell size [48] in the case of MAPI, which was attributed to the highly screened nature of perovskites. It is notable that the Lany–Zunger correction (multiplying Makov–Payne $E^{q}$ by 0.65) provides even better convergence versus the supercell size; therefore, we used it in our calculations where appropriate. Contrary to hybrid perovskites, the dielectric constant of Cs-based perovskites does not show a strong temperature-dependence [49]; as such, we used the measured effective dielectric constants of 8.6 [50] for CsPbI_2_Br and 10 for α-CsPbI_3_ [51]. In the text any time we mention an atom X in a chemical formula we imply a halogen I or Br. 

## 3. Results and Discussion

### 3.1. Lattice Constants

To test the validity of our methodology against experimental data, we calculated the lattice constants of cubic and tetragonal-phase CsPbI_3_ and CsPbI_2_Br. For consistency, we used normalized lattice constants as in reference [52]; in the case of tetragonal phases, this was done by rotating the coordinate system in such a way that the a and b vectors were aligned along the Pb-X bonds. The lattice constants we calculated for α-CsPbI_3_ and β-CsPbI_3_ were generally in good agreement with experimental data and previous calculations (Table 1). In the case of pseudocubic α-CsPbI_2_Br, we used α-CsPbI_3_ $Pm\bar{3}m$ as the starting point; that is, replacing an iodine with a bromine yielded α-CsPbI_2_Br *P4/mmm*, which, according to our calculations, was the most stable pseudocubic phase. 

By doubling the α-CsPbI_3_ $Pm\bar{3}m$ cell along all three directions, we were able to take into consideration more possible α-CsPbI_2_Br phases. *Inma* was equally stable to *P4/mmm* and was then energetically followed by *Cm* and *Pmm2* (Figure 1). To obtain the “clusterized” *C1* phase, we considered a 3 × 3 × 3 supercell and made I -> Br substitutions in a localized section of the supercell (Figure 2), and the least stable α-CsPbI_2_Br phase was *Amm2*. It is notable that *a = b* ≠
*c* in the case of α-CsPbI_2_Br as the Pb-Br bond lengths were shorter than the Pb-I ones and thus a truly cubic phase was not possible to maintain. The main feature which distinguished pseudocubic α-CsPbI_2_Br from β-CsPbI_2_Br was that in the latter, the PbX_6_ octahedra were rotated with respect to one another. Octahedral rotations are associated with soft phonon modes [53] and these modes, along with their anharmonic behavior [52], are responsible for black phase instability in CsPbI_3_ at lower temperatures [54]. It is interesting to note that for pseudocubic structures, the average lattice parameter was nearly the same for all phases and only the spread of the values greatly differed. Notably, the *I4/mcm* phase had the largest spread with *a = b =* 6.31 Å and *c =* 5.93 Å.

In the case of β-CsPbI_2_Br, we started with β-CsPbI_3_ *I4/mcm* and substituted iodines for bromines in a variety of ways (Figure 3). Due to the presence of octahedral tilting, it was possible to define apical and equatorial substitutions in the cell, which corresponded to *I4/mcm* and *Fmmm*, respectively. The latter has previously [55] been predicted to be more stable by 58 meV per formula unit, and our results were in concordance with their result. Thus, our results indicated that *Fmmm* was the most stable phase and *I4/mcm* was the least stable by 70 meV/f.u. Meanwhile, the “mixed” low-symmetry *C*2 phase was energetically in between them and 40 meV/f.u. less stable than *Fmmm*.

**Table 1 nanomaterials-13-00276-t001:** Normalized lattice constants (Å) of various experimentally measured and theoretically simulated α-CsPbI3 β-CsPbI3, α-CsPbI2Br and β-CsPbI2Br phases. Values obtained in this work are in italics, values inside of round parentheses were computed with spin–orbit coupling included. Note that for α phases, only the a lattice parameter is represented while for β phases, the a and b parameters are shown.

	**Phase**	**Lattice Parameter**		**Energy/f.u.** **(eV/cell)**
		**a**		
**Exp**	α-CsPbI_3_	6.29 [31], 6.297 [52]		-
DFT	α-CsPbI_3_ $\mathrm{Pm}\bar{3}m$	6.18 [52], 6.26 [56], 6.27 (6.30), 6.41 [57]		**−15.28**
**Exp**	α-CsPbI_2_Br	6.138 [30]		−
DFT	α-CsPbI_2_Br P4/mmm	6.18 (6.17) ± 0.22, 6.26 [29]		**−15.84**
	α-CsPbI_2_Br C1	6.19 ± 0.02		−15.82
	α-CsPbI_2_Br Inma	6.18 ± 0.05		**−15.84**
	α-CsPbI_2_Br Cm	6.18 ± 0.13		−15.83
	α-CsPbI_2_Br Pmm2	6.19 ± 0.04		−15.82
	α-CsPbI_2_Br Amm2	6.19 ± 0.05		−15.80
	**Phase**	**Lattice Parameters**		**Energy/f.u.** **(eV/cell)**
		**a**	**b**	
**Exp**	β-CsPbI_3_	6.241 [52]	6.299 [52]	-
DFT	β-CsPbI_3_ I4/mcm	5.97 [52], 6.28	6.273 [52], 6.33	**−15.30**
**Exp**	β-CsPbI_2_Br	6.130 [30]	6.088 [30]	-
DFT	β-CsPbI_2_Br I4/mcm	6.33 [29], 6.29	6.03 [29], 5.96	−15.84
	β-CsPbI_2_Br Fmmm	6.08 [55], 6.01 (6.03)	6.45 [55], 6.38 (6.37)	**−15.92**
	β-CsPbI_2_Br C2	6.14	6.19	−15.87

To compare the most stable mixed alpha and beta phases in more detail, we calculated the deviation of α-CsPbI_2_Br and β-CsPbI_2_Br from Vegard’s law. This was done by linearly projecting the lattice constants versus the Br concentration with the appropriately phased CsPbI_3_ and CsPbBr_3_ as the “end points”. In the case of α-CsPbI_2_Br, we noted a small average Vegard’s deviation of just −0.02 Å. We also calculated the deviation with spin–orbit coupling (SOC) included and found that it was −0.01 Å. In the case of β-CsPbI_2_Br *Fmmm,* the deviation was much larger, being 0.14 Å for *a = b* and −0.16 Å for *c*. This deviation changed to 0.11 Å and −0.18 Å, respectively, when SOC was turned on. The limited effected of SOC on perovskite geometry has previously been observed in other works [46]. These deviations mean that the c-axis is too elongated when compared to Vegard’s law, which implies a weakening of the Pb-I apical bond, whereas the equatorial plane is much more compact than one would expect, implying a stronger bonding in this plane. When analyzing the geometrical differences of β-CsPbI_2_Br *Fmmm* with the respective pristine tetragonal phases of β-CsPbI_3_ and β-CsPbBr_3_ *I4/mcm*, we noted that the octahedral rotations were much larger for β-CsPbI_2_Br *Fmmm* than for β-CsPbI_3_ or β-CsPbBr_3_ *I4/mcm*. Moreover, the Pb-I bond length on the apical positions was 0.03 Å longer than for β-CsPbI_3_ *I4/mcm*. The major deviation from Vegard’s law can be noted via visual inspection in Figure 4, where we compare *Fmmm* to the corresponding CsPbI_3_ and CsPbI_3_ β-phases when looking down the c-axis. The differences were quite notable: the latter two had an equatorial Pb-X-Pb angle of 165.8° whereas the former decreased to 157.9° showing a significantly larger rotation of the PbX_6_ octahedra. Finally, we noted that the rotation of α-CsPbI_2_Br *P4/mmm* into β-CsPbI_2_Br *I4/mcm* energetically cost practically nothing since our method predicted only 0.002 eV/halide anion difference; it appears that the same was true for a transformation of α-CsPbI_2_Br *P4/mmm* into β-CsPbI_2_Br *Fmmm*; thus, it is questionable if such a phase shift can be selective. 

### 3.2. Mechanical Properties

Given the curious geometrical trends observed in the previous section, we considered the bulk modulus of α-$CsPbI_{x}Br_{3-x}\;\left(x=0,\;1,\;3\right)$ by way of calculating the energetic response to strain and fitting the results to the Birch–Murnaghan equation of state (EOS), shown below in Equation (6) in the E vs. V formulation:(6)$$E\left(V\right)=E_{\circ }+\frac{9\times B_{\circ }V_{\circ }}{16}\left({\left({\left(\frac{V_{\circ }}{V}\right)}^{\frac{2}{3}}-1\right)}^{3}\times B_{\circ }^{\prime }+\left({\left({\left(\frac{V_{\circ }}{V}\right)}^{\frac{2}{3}}-1\right)}^{2}\times \left(6-4\times {\left(\frac{V_{\circ }}{V}\right)}^{\frac{2}{3}}\right)\right)\right)$$
where E is the energy, $E_{\circ }$ is the equilibrium energy, V is the volume, $V_{\circ }$ is the equilibrium volume, and $B_{\circ }$ and $B_{\circ }^{\prime }$ refer, respectively, to the invariant and linear response of the bulk modulus; B is related to pressure P, as described below in Equation (7):(7)$$B=B_{\circ }+B_{\circ }^{\prime }\times P$$
where P is typically taken to be 1 atm. The results (Appendix A) tabulated in Table 2 show that a higher I concentration resulted in a lower bulk modulus. The ambient bulk modulus ($B_{\circ }^{\prime })$ was relatively proportionate to the amount of Br; a one-third doping ratio of Br resulted in around 27% of the overall increase of $B_{\circ }^{\prime }$. Previously, contrasting results were found for MAPbX_3_ (X = Br, I) where the iodine-based perovskite exhibited a 1.9 GPa higher bulk modulus at 15.0 GPa when compared to the bromine-based perovskite; that implied that the A cation size significantly influenced the trend [58]. For the case of Cs in the A sites, the bulk modulus was as high as 18.0 GPa for the I in the X site, which was 4.3 GPa lower than in the Br case. This is in line with other studies also employing the Birch–Murnaghan EOS, which found $B_{\circ }=20.7\;\mathrm{GPa}$ and $B_{\circ }^{\prime }=4.881$ for cubic CsPbBr_3_ [59] ($B_{\circ }=9.618$ for the surface of CsPbBr_3_ [60]), and $B_{\circ }=17.82\;\mathrm{GPa}$ and $B_{\circ }^{\prime }=4.665$ for cubic CsPbI_3_ [61]. The examination of the invariant and linear response variables showed that higher iodine concentrations resulted in a higher pressure-invariant component of the bulk modulus; however, notably, the pure I and Br cases had comparable linear response components, which were both nearly 0.2 lower than those for CsPbI_2_Br. While the effect at ambient pressures was not enough to overcome that of the invariant term, these calculations indicate that at higher pressures, the mixed halogens will exhibit a lower strain response than the pure perovskites.

### 3.3. Mixing Energy

Since the free-energy of cubic $\mathrm{CsPb}{\left(I_{1-x}{\mathrm{Br}}_{x}\right)}_{3}$ was investigated quite thoroughly previously [34], we focused our attention, as in the lattice constant section, on the distinctions between α-CsPbI2Br and β-${\mathrm{CsPbI}}_{2}\mathrm{Br}$. 

The mixing energy is defined as:(8)$$\Delta U_{M}\left(\mathrm{CsPb}{\left(I_{1-x}{\mathrm{Br}}_{x}\right)}_{3}\right)=\frac{E\left(\mathrm{CsPb}{\left(I_{1-x}{\mathrm{Br}}_{x}\right)}_{3}\right)-\left(1-x\right)E\left({\mathrm{CsPbI}}_{3}\right)-x*E\left({\mathrm{CsPbBr}}_{3}\right)}{N}$$
where $E\left(\mathrm{CsPb}{\left(I_{1-x}{\mathrm{Br}}_{x}\right)}_{3}\right)$ is the energy of a $\mathrm{CsPb}{\left(I_{1-x}{\mathrm{Br}}_{x}\right)}_{3}$ configuration and E(${\mathrm{CsPbI}}_{3}$) and E(${\mathrm{CsPbBr}}_{3}$) are the energies of pure cesium–lead–iodide and cesium–lead–bromide perovskites, respectively, in the appropriate phases; N is the number of anion halides per ${\mathrm{CsPbI}}_{2}\mathrm{Br}$ cell. We chose to use this unit to keep our units consistent with previous results [62] for MAPbBr_x_I_(1−x)_. Figure 5a presents $\Delta U_{M}$ for all the phases considered in Section 3.1. In the case of both the alpha and beta phases, we used the same phases of pure perovskites when calculating the mixing energy. Generally, we noted a significantly higher mixing energy of α-$\mathrm{CsPb}{\left(I_{1-x}{\mathrm{Br}}_{x}\right)}_{3}$, which indicated that β-$\mathrm{CsPb}{\left(I_{1-x}{\mathrm{Br}}_{x}\right)}_{3}$ would generally have better miscibility for all values of x.

When considering the effects of configurational entropy, we refer to Section 3.1 and note that the α-${\mathrm{CsPbI}}_{2}\mathrm{Br}$ in fact had two dominant motifs in its unit cells: one with three halide sites and another with six, whereas, for β-${\mathrm{CsPbI}}_{2}\mathrm{Br}$, only twelve appeared. Therefore, the difference in stability was even greater than Figure 5a would suggest; we plotted these contributions in Figure 5b. It is natural to question that perhaps it could be possible to stabilize this mixed perovskite by modifying the experimental conditions to selectively prepare β-${\mathrm{CsPbI}}_{2}\mathrm{Br}$, which would be much less miscible; however, as we show in the next section, the redox chemistry is equally important to consider before making any such conclusions.

### 3.4. Redox Properties of α-CsPbI_2_Br and β-CsPbI_2_Br

In this section, we consider the redox properties of α-CsPbI_3_, α-CsPbI_2_Br *P4/mmm*, α-CsPbI_2_Br *C1,* β-CsPbI_2_Br *I4/mcm* and β-CsPbI_2_Br *Fmmm*. By applying Equation (4), one may judge the oxidative capability of the materials. For the sake of consistency, we list them at E_f_ = 0 eV for oxidation and E_f_ = E_g_ for reduction (Table 3). Just as in the case of the molecular I_2_ and Br_2_ species, there was a very notable difference between CsPbI_3_ and CsPbBr_3_: the latter was significantly harder to oxidize at E_F_ = 0 eV. Surprisingly, the oxidative capability of CsPbI_2_Br differed very little from that of CsPbI_3_ in all phases except for β-CsPbI_2_Br *Fmmm*, which was more resilient than the other mixed phases. Adding an electrostatic correction did not essentially change the patterns observed. For completeness, we also list the Fermi pinning energy of the pristine material and its energy of oxidation (or reduction since they are equal at the pinning E_F_). We note that such donors and acceptors may also couple to band-edge carriers; thus, the quantities considered in Table 3 are those relating ed to free carriers (independent of chemical potentials) [63]. The fact that the energy of oxidation of the pristine material did not change very much by a doping of 33% of bromine was somewhat unexpected. This is apparently related to the fact that the iodine and bromine atoms are well separated near valence-p-contributions in the partial density of states (DOS) (Figure 6, Appendix A). Thus, the iodine sites are oxidized preferentially over the bromine sites. An increase in the bromine content causes the spectroscopic Br *p*-contribution to grow, but at the same time the bandgap also widens, and oxidation of the mixed material ought to become less favorable. Thus, two phases emerge; namely, the Br-rich mixed perovskite bulk and iodine-rich nanoinclusions; the latter outcompetes the former in the generation of carriers by virtue of having a smaller E_g_. This simple description is in line with the unified theory of light segregation [27].

We note that this line of reasoning only applies to the well-mixed phases of the material. Higher bromine content does not, in fact, always lead to a larger gap. The calculated band gap of α-CsPbI_3_ was 1.362 eV whereas α-CsPbI_2_Br *P4/mmm* had a somewhat larger bandgap of 1.402 eV, and the α-CsPbI_2_Br *C1 cluster type* (Figure 2) had a bandgap of just 1.311 eV. This can be understood by the observation that the Br-rich cluster inclusion strains the lattice and forces the Pb-I bond to be shorter than in *P4/mmm*, where it is allowed to relax. The bond constraint increases the VBM energy due to its antibonding (I 5*p*–Pb 6*s*) nature [64]. Our calculations confirmed this, since only the VBM shifted from 1.550 eV for *P4/mmm* to 1.657 eV for the *C1* cluster type whereas the conduction band maximum (CBM) remained static. The latter was, however, still lower than the VBM of 1.765 eV calculated for α-CsPbI_3_ $Pm\bar{3}m.\;$ All of the values for CsPbI_2_Br somewhat underestimated the experimental value of 1.92 eV for CsPbI_2_Br [65,66,67]. Our results indicated that Br-rich nanoinclusions ought to have the effect of facilitating carrier generation; however, this was likely balanced by the increased strain it applied to the lattice. Despite the soft nature of the material, the experimentally observed interface formation [23] confirmed the tendency to release this strain via segregation. It is notable that in both the cases of α-${\mathrm{CsPbIBr}}_{2}$ *P4/mmm* and *C1*, the E_VBM_ was higher than in α-CsPbI_3_, which means that in a hypothetical photovoltaic device they would both donate holes to α-CsPbI_3_. When analyzing the beta phases, we noted that these materials had a predicted bandgap which was somewhat wider than in the alpha-phase materials. Overall, the pristine pinning energy did not change much at all for a doping Br concentration of 33% in all cases, and the oxidation energy at the Fermi pinning energy was extremely similar for all the materials analyzed, even including α-CsPbBr_3_ (Table 3).

To further analyze the redox chemistry of these materials, we calculated the CFD photooxidation reaction presented in Equation (1) by considering this reaction as a reaction of a hole-coupled iodide oxidation between a pure 3 × 3 × 3 slab and a pure charged 3 × 3 × 3 slab resulting in the neutral slab receiving an interstitial iodine and the charged slab gaining a halide vacancy. This process is described by the following equation:(9)$$E_{\mathrm{ox}}\left(I_{I}^{x}+h^{•}\right)=E\left(I_{I}^{x}\right)+E\left(V_{X}^{•}\right)-\left[E{\left({\mathrm{CsPbI}}_{2}\mathrm{Br}\right)}_{\mathrm{pure}}+E{\left({\mathrm{CsPbI}}_{2}{\mathrm{Br}}^{+}\right)}_{\mathrm{pure}}\right]$$

It is notable that the mobility of the halide vacancies allows for a halide exchange such as
(10)$$I_{I}^{x}+h^{•}\to {\mathrm{Br}}_{i}^{x}+V_{I}^{•}$$

There are in fact four permutations for CFDs to consider in Equation (10); namely, $I_{i}^{x}/V_{I}^{•}$, $I_{i}^{x}/V_{\mathrm{Br}}^{•}$,${\mathrm{Br}}_{i}^{x}/V_{\mathrm{Br}}^{•}$ and ${\mathrm{Br}}_{i}^{x}/V_{I}^{•}$. Two other relevant possibilities are also to be considered: the possibility of hole-free oxidation into a neutral Frenkel defect pair $X_{i}^{\prime }/V_{X}^{•}$ or $X_{i}^{x}/V_{X}^{x}$ and a Frenkel defect pair $X_{i}^{•}/V_{I}^{•}$ in a hole-rich regime. All of these possibilities were explored, and the results of the calculations are summarized in Table 4. 

It can be seen from these results that α-CsPbI_2_Br *P4/mmm* only showed instability towards hole-free CFDs involving bromine, either with bromine or iodine vacancies, indicating a lack of thermodynamic preference for vacancy type. Meanwhile, the α-CsPbI_2_Br *C1* phase showed instability towards the hole-free $I_{i}^{\prime }/V_{I}^{•}$ CFD, further supporting the destabilizing strain effects a bromine-rich nanoinclusion has on the surrounding CsPbI_3_; however, it appears that the inclusion was also somewhat destabilized since the hole-free ${\mathrm{Br}}_{i}^{\prime }$/$V_{I}^{•}$ CFD pair also had a low E_ox_. In the case of all α-CsPbI_2_Br phases, we noted that only hole-coupling and two-hole-coupling CFD generation was favorable. In the case of the α-CsPbI_2_Br *C1* phase, the hole-coupled and two-hole-coupled $I_{i}^{x}/V_{I}^{•}$ pairs were also much more destabilized when compared to α-CsPbI_2_Br *P4/mmm*.

One mechanism by which the phase transformation of an a-phase perovskite to a b-phase perovskite can take place is a defect-coupled process where Pb*X*_6_ octahedra are allowed to release lattice strain through rotation in relation to one another (Figure 4). To test this hypothesis, we considered the α-CsPbI_2_Br *P4/mmm* phase of the material (Table 4), but with the coordinate system rotated by 45° around the *c*-axis, thus aligning its coordinate system with that of the β-CsPbI_2_Br phases. The hypothesis was confirmed since the material retained its character in the defect-free supercell, but as soon as vacancies or interstitials were introduced there was a clear rotation of the octahedra, and a *b*-phase emerged. Thus, our results strongly support this hypothesis and suggest that α-CsPbI_2_Br may be unstable not just because it can segregate, but also because it will transform into β-CsPbI_2_Br upon accumulation of CFD defects. We note the extremely favorable nature of CFD formation in the rotated α-CsPbI_2_Br P4/mmm phase is indicative of the thermodynamic favorability of this process.

In the case of β-CsPbI_2_Br, we observed a very different behavior where the equatorial *Fmmm* phase was stable towards all pathways of oxidation considered and the apical one was highly unstable towards all mechanisms involving either charged defects or hole coupling. This indicates that the *Fmmm* phase is not only much more resilient towards segregating into CsPbI_3_/CsPbBr_3_, but it will also generate far fewer CFDs upon light exposure. We conclude this section with a DFT-based prediction that the selective preparation of β-CsPbI_2_Br *Fmmm* should yield a significantly more stable mixed material; however, if one assumes that it occurs from the defect-coupled rotations of octahedra in α-${\mathrm{CsPbIBr}}_{2}$ *P4/mmm*, then it is unlikely that β-CsPbI_2_Br *Fmmm* could be easily and selectively prepared. A visual analysis of the results of our defect calculations confirmed as much (Figure 7 and Appendix A). In conclusion, we stress the most important result of this section, which is that β-CsPbI_2_Br *Fmmm* has photostability far superior to any phase of α-CsPbI_2_Br and, surprisingly, it is even superior to α-CsPbI_3_. The engineering of such a highly stable phase might be combined with other strategies such as nonstoichiometry [68] or the release of microstrain [69] to achieve a superior perovskite-based photovoltaic material. 

## 4. Conclusions

The straightforward logic of why one ought to mix the superior photovoltaic properties of CsPbI_3_ with the increased stability of CsPbBr_3_ seems quite at odds with how difficult it is to understand why such materials are unstable. Even a conservative amount of bromine additive greatly changes the photochemistry of the material. Bromine-rich nanoinclusions of CsPbBr_3_ into α-${\mathrm{CsPbI}}_{2}\mathrm{Br}$ leave it more susceptible to oxidation via charged Frenkel defect pairs. Even worse, the formation of these charged vacancy and interstitial defect pairs outright destabilizes the pseudocubic phase and allows the PbX_6_ octahedra to rotate into β-CsPbI_2_Br. Further complications arise from the fact that β-CsPbI_2_Br has two phases which are quite different in their resilience against CFD formation and photo segregation. In the present study, we found that the equatorial phase appeared to be very stable as a mixed solid solution and was resilient against CFD formation, whereas the apical phase was highly vulnerable to CFD formation and had a much higher mixing energy. This major difference in mixing energies implies that the material is likely capable of assuming the equatorial phase selectively, which means that selectively isolating the phase is a very real possibility in expanding the material scientist’s toolbox in engineering a stable perovskite photovoltaic cell. Given that the band gap seems to be somewhat wider for this phase than for the α-${\mathrm{CsPbI}}_{2}\mathrm{Br}$ phase, we have to conclude that the equatorial β-CsPbI_2_Br is ultimately a compromise of some photovoltaic performance in favor of photostability. An additional potential difficulty is seen in the mechanism by which theory predicts the formation of such a phase: it originates from the rotations of Pb*X*_6_ octahedra in α-${\mathrm{CsPbI}}_{2}\mathrm{Br}$, and thus a complex scheme might be necessary for successfully isolating this phase. Our calculations suggest that an experimental synthesis of the promising equatorial β-CsPbI_2_Br should probably bypass all α-${\mathrm{CsPbI}}_{2}\mathrm{Br}$ phases to avoid random rotations into various β-CsPbI_2_Br phases.

## Figures and Tables

**Figure 1 nanomaterials-13-00276-f001:**
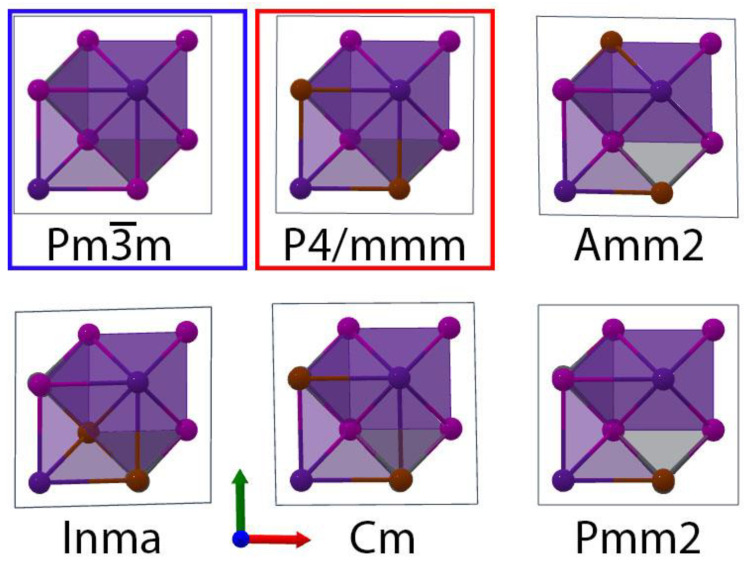
The crystal structures of the α-CsPbI_3_ $Pm\bar{3}m$ (blue box) and α-CsPbI_2_Br P4/*mmm* (red box) phases, and all other α-CsPbI_2_Br phases considered. The perspective is down the *c*-axis of the crystals with the lead atoms eclipsed by halides to better represent the halide substitutions.

**Figure 2 nanomaterials-13-00276-f002:**
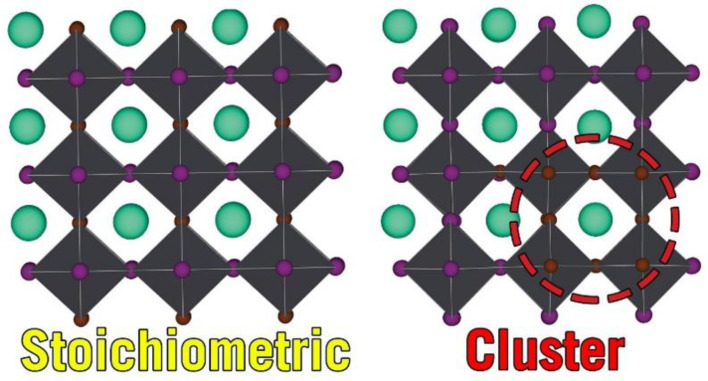
The crystal 3 × 3 × 3 supercells of the α-CsPbI_2_Br *P4/mmm* (**left**) and α-CsPbI_2_Br *C1* (**right**) phases.

**Figure 3 nanomaterials-13-00276-f003:**
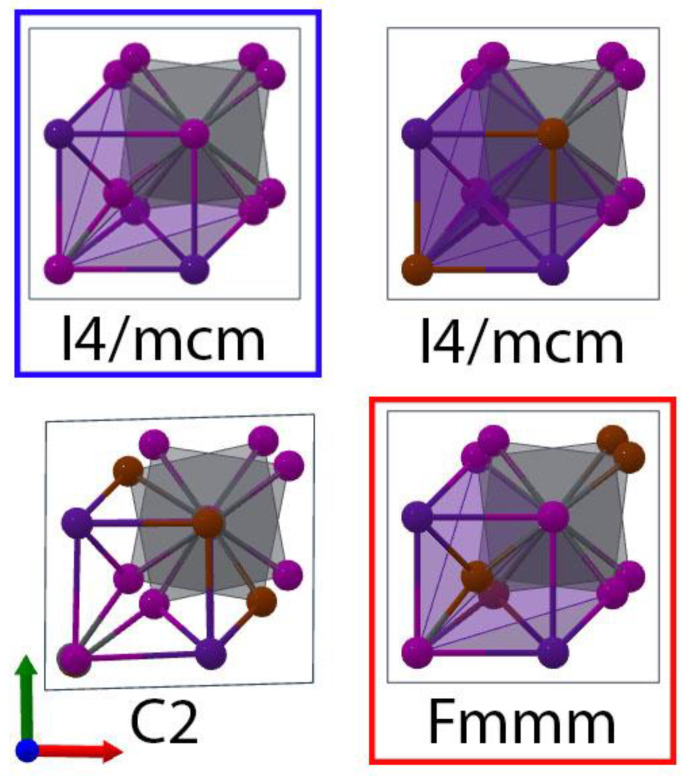
The crystal structures of the β-CsPbI_3_ *I4/mcm* (blue box) and β-CsPbI_2_Br F*mmm* (red box) phases and all other β-CsPbI_2_Br phases considered. The perspective is down the *c*-axis of the crystals with the lead atoms eclipsed by halides to better represent the halide substitutions. In this case, the system of coordinates was also rotated so that a and b became aligned with Pb-X bonds for easy comparison to Figure 1.

**Figure 4 nanomaterials-13-00276-f004:**
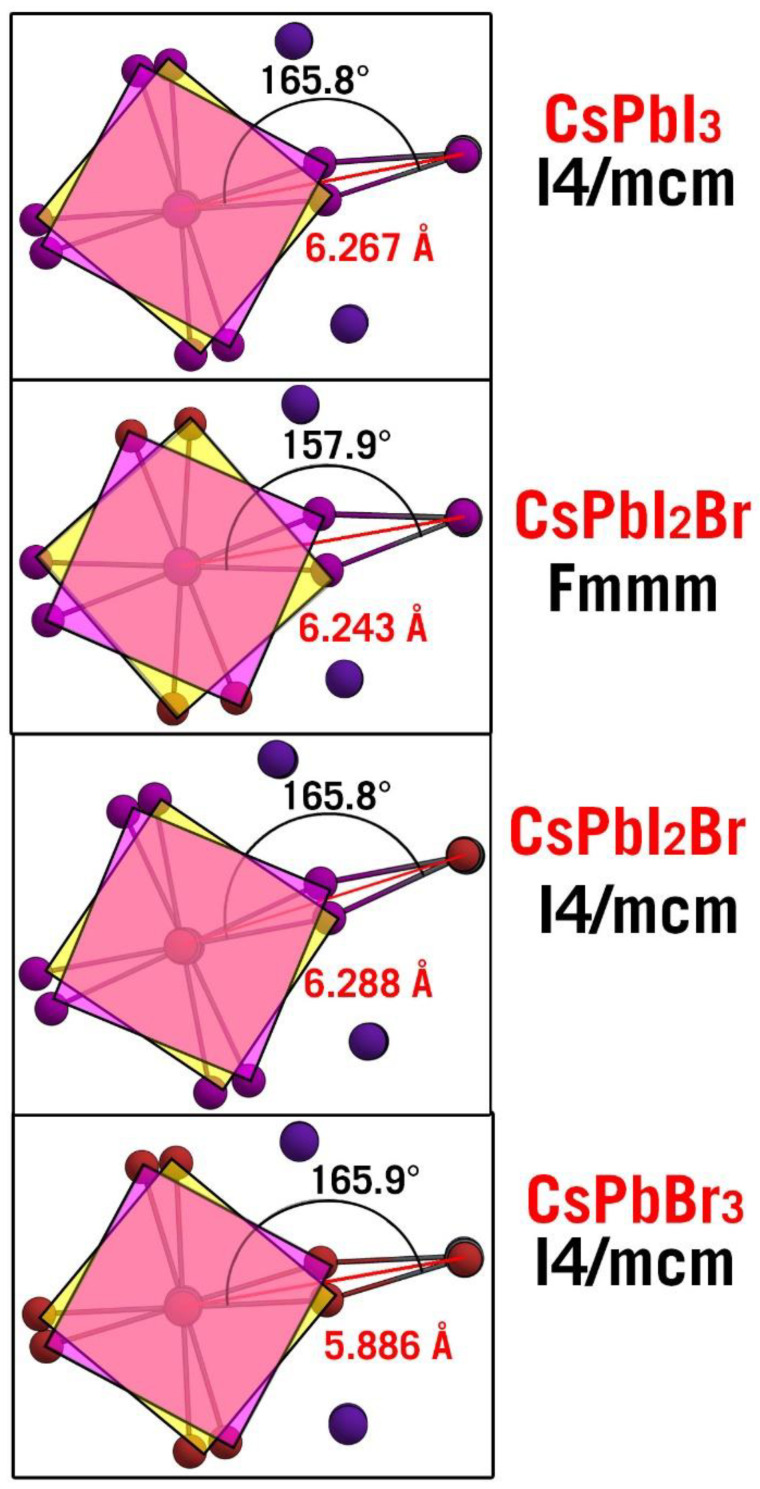
A view down the c-axis of the β phases of CsPbI_3_, CsPbI_2_Br (*I4/mcm* and *Fmmm*) and CsPbBr_3_. The colored squares indicate the equatorial planes of the PbX_6_ octahedra for easier comparison.

**Figure 5 nanomaterials-13-00276-f005:**
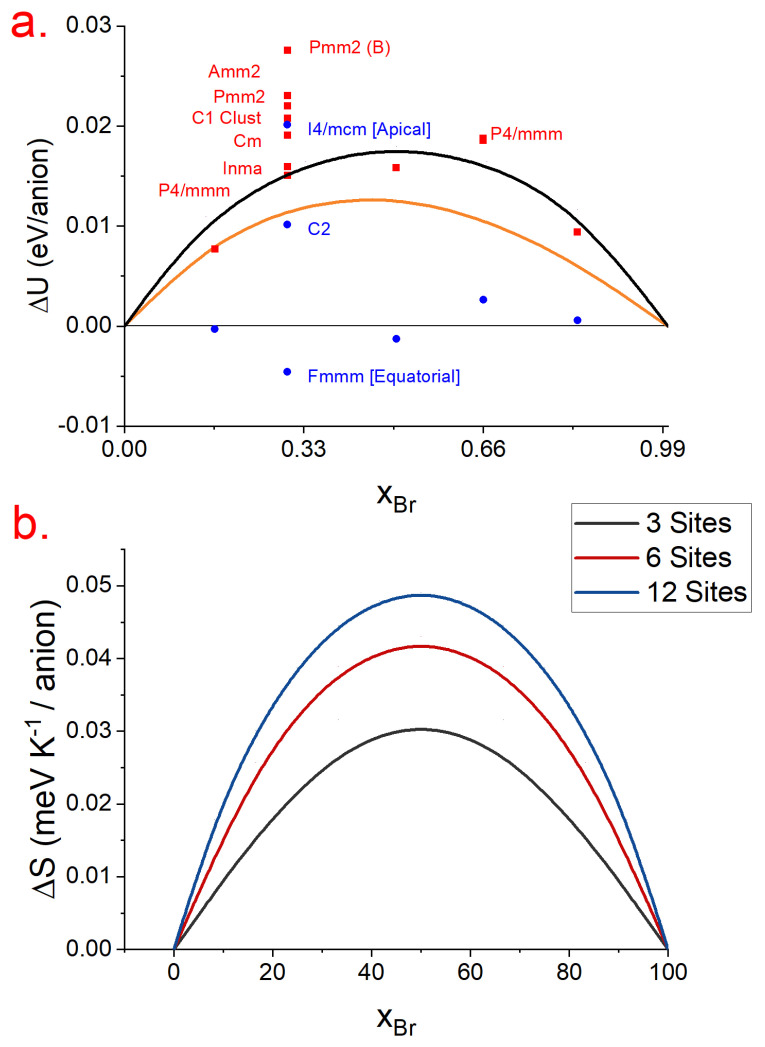
(**a**) The mixing energies $\Delta U_{M}$ of all the phases calculated. The room temperature solid solution approximation Ωx(1 − x) Ω = 0.06 − 0.02x (orange line) and the approximation for a completely random alloy in the high temperature limit (black line) are also plotted; (**b**) The configurational entropy depending on the number of halide sites in the unit cell.

**Figure 6 nanomaterials-13-00276-f006:**
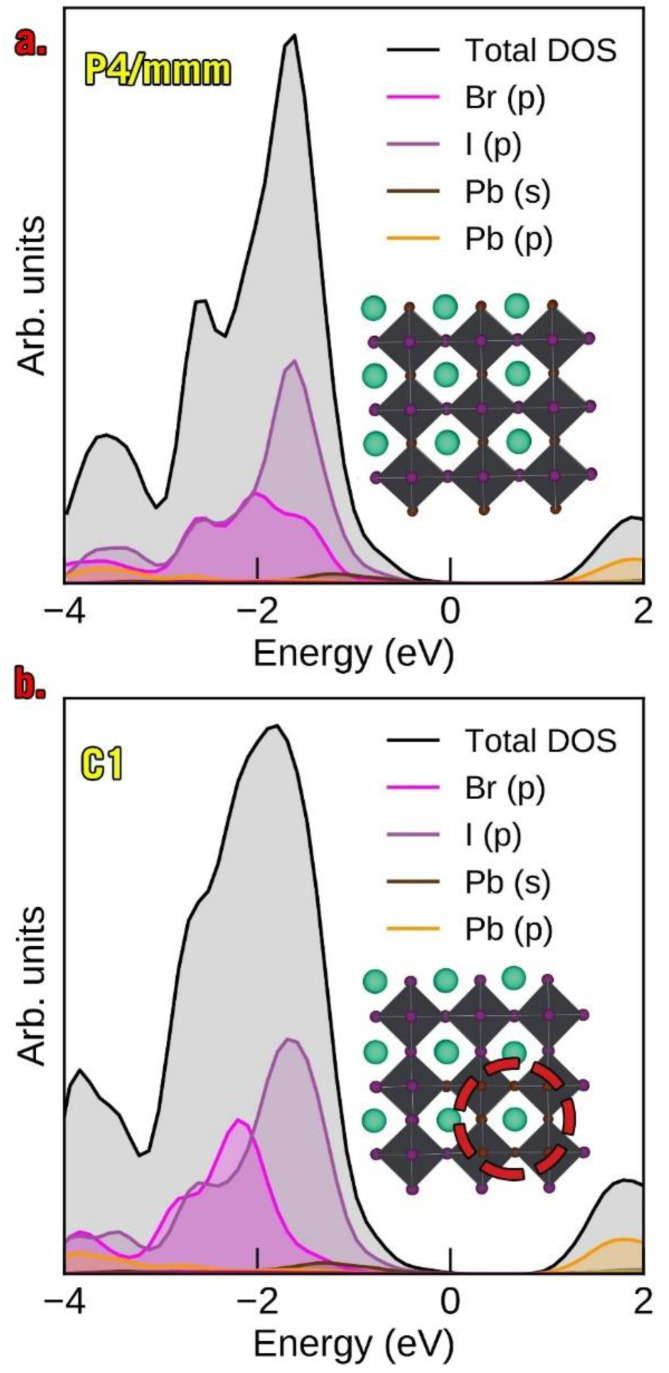
The partial density of states of (**a**). α-CsPbI_3_ P4/*mmm* and (**b**). α-CsPbI_3_ *C1* (cluster type).

**Figure 7 nanomaterials-13-00276-f007:**
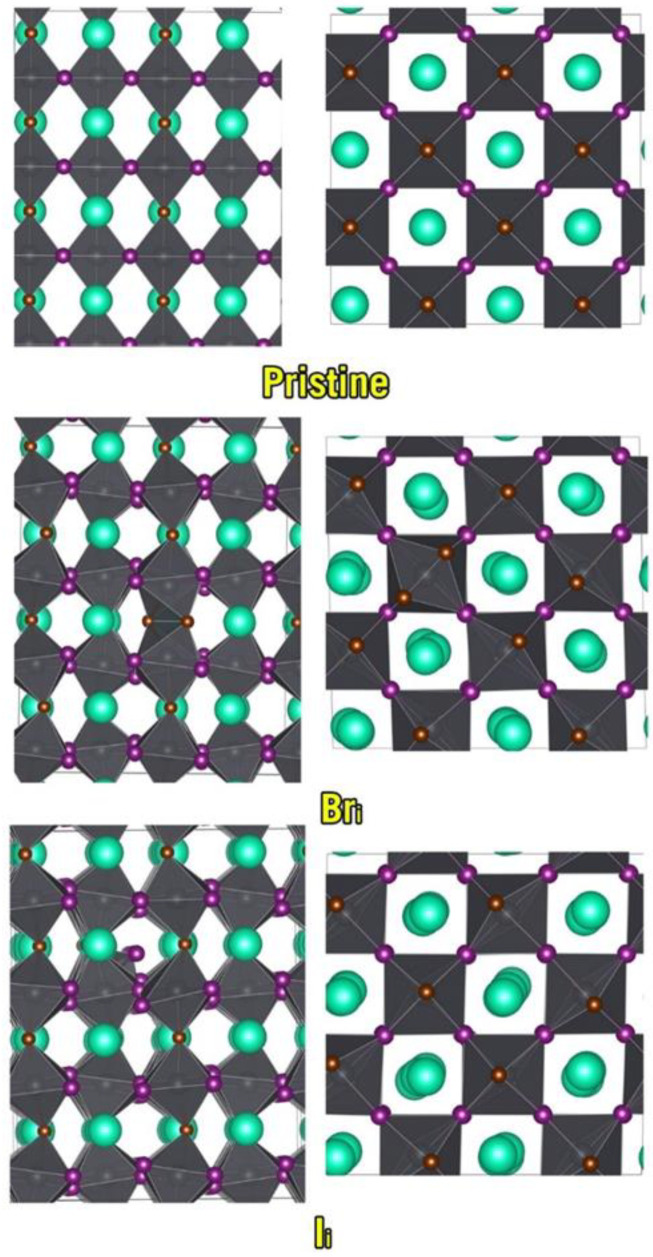
α-CsPbI_2_Br *P4/mmm* with a rotated coordinate system to align with β-CsPbI_2_Br and the same lattice with iodine and bromine interstitials.

**Table 2 nanomaterials-13-00276-t002:** Fitted zero- ($B_{\circ }$) and first-order ($B_{\circ }^{\prime }$, dimensionless) portions of the bulk modulus (B) from the energy vs. volume response curve fit to the Birch–Murnaghan equation of state.

	$B_{\circ }\;\left(GPa\right)$	$B_{\circ }^{\prime }$	$R^{2}$
α-CsPbBr_3_ $Pm\bar{3}m$	22.31	4.67	0.99997
α-CsPbI_2_Br *P4/mmm*	19.20	4.85	1.00000
α-CsPbI_3_ $Pm\bar{3}m$	18.04	4.67	0.99996

**Table 3 nanomaterials-13-00276-t003:** Energies of oxidation and reduction at E_F_ = 0 and E_F_ = E_g_, respectively, as well as the energy of oxidation or reduction at the pristine Fermi pinning energy E_pin_*. For the former three, we included Lany–Zunger corrected values in parentheses. The valence band maximum EVBM and direct band gap energy are also included. The * distinguishes these oxidation energies from the Frenkel oxidation energies of future tables.

	E_ox_ * $\;(I_{I}^{x}+h^{•}\to I_{I}^{•})\;@EF_{\;}=0\;(\mathrm{eV})$	E_red_ * $(I_{I}^{x}+e^{\prime }\to I_{I}^{\prime })\;@\;$ E_F_ = E_g_ (eV)	E_redox_ * $\left(I_{I}^{x}+h^{•}\to I_{I}^{•}\right)$ @ E_F_ = E_pin_ (eV)	E_pin_ * (eV)	E_VBM_ (eV)	E_g_ (eV)
α-CsPbI_3_ $Pm\bar{3}m$	0.25 (0.34)	0.08 (0.17)	0.85 (0.94)	0.60	1.765	1.362
$\alpha -{\mathrm{CsPbIBr}}_{2}$ *P4/mmm*	0.25 (0.36)	0.11 (0.22)	0.85 (0.99)	0.63	1.550	1.402
$\alpha -{\mathrm{CsPbIBr}}_{2}$ *C1*	0.26 (0.36)	0.08 (0.19)	0.89 (0.93)	0.57	1.657	1.311
$\;\beta -{\mathrm{CsPbIBr}}_{2}$ *Fmmm* (Equatorial)	0.17 (0.28)	0.16 (0.27)	0.93 (1.04)	0.75	1.440	1.526
$\;\beta -{\mathrm{CsPbIBr}}_{2}$ *I4/mcm* (Apical)	0.24 (0.35)	−0.01 (0.09)	0.85 (0.96)	0.61	1.475	1.539
α-CsPbBr_3_ $Pm\bar{3}m$	0.83 (0.93)	−0.02(0.06)	1.01 (1.11)	0.17	1.212	1.606

**Table 4 nanomaterials-13-00276-t004:** Energies of the neutral, hole-coupled and bihole-coupled Frenkel oxidations $E_{\mathrm{ox}}$ for α -CsPbI_3_ $Pm\bar{3}m$, α-CsPbI_2_Br *P4/mmm* (stochastic), α-CsPbI_2_Br *C1* (cluster), α-CsPbI_2_Br *P4/mmm* with the coordinate system rotated by 45°, β-CsPbI_2_Br *Fmmm* (equatorial) and I4/mcm (apical). We indicate negative or near-zero values with ***bold*** font.

	Redox Reaction:	$E_{ox}\left(I_{I}^{x}\right)$ (eV)	Redox Reaction	$E_{ox}\left(I_{I}^{x}+xh^{•}\right)$ (eV)
$\alpha -{\mathrm{CsPbI}}_{3}\;\mathrm{Pm}\bar{3}m$	$I_{I}^{x}\to I_{i}^{x}+V_{I}^{x}$ $I_{I}^{x}\to I_{i}^{\prime }+V_{I}^{•}$	2.84 1.08	$I_{I}^{x}+h^{•}\to I_{i}^{x}+V_{I}^{•}$ $I_{I}^{x}+2h^{•}\to I_{i}^{•}+V_{I}^{•}$	1.25 0.63
α-CsPbI_2_Br P4/mmm	$I_{I}^{x}\to I_{i}^{x}+V_{I}^{x}$ $I_{I}^{x}\to I_{i}^{\prime }+V_{I}^{•}$	2.92 0.97	$I_{I}^{x}+h^{•}\to I_{i}^{x}+V_{I}^{•}$ $I_{I}^{x}+2h^{•}\to I_{i}^{•}+V_{I}^{•}$	1.21 0.46
	${\mathrm{Br}}_{\mathrm{Br}}^{x}\to {\mathrm{Br}}_{i}^{x}+V_{\mathrm{Br}}^{x}$ ${\mathrm{Br}}_{\mathrm{Br}}^{x}\to {\mathrm{Br}}_{i}^{\prime }+V_{\mathrm{Br}}^{•}$	1.65 **−0.18**	${\mathrm{Br}}_{\mathrm{Br}}^{x}+h^{•}\to {\mathrm{Br}}_{i}^{x}+V_{\mathrm{Br}}^{•}$ ${\mathrm{Br}}_{\mathrm{Br}}^{x}+2h^{•}\to {\mathrm{Br}}_{i}^{•}+V_{\mathrm{Br}}^{•}$	**−0.02** **−0.01**
	$I_{I}^{x}\to I_{i}^{x}+V_{\mathrm{Br}}^{x}$ $I_{I}^{x}\to I_{i}^{\prime }+V_{\mathrm{Br}}^{•}$	2.98 1.07	$I_{I}^{x}+h^{•}\to I_{i}^{x}+V_{\mathrm{Br}}^{•}$ $I_{I}^{x}+2h^{•}\to I_{i}^{•}+V_{\mathrm{Br}}^{•}$	1.31 0.56
	${\mathrm{Br}}_{\mathrm{Br}}^{x}\to {\mathrm{Br}}_{i}^{x}+V_{I}^{x}$ ${\mathrm{Br}}_{\mathrm{Br}}^{x}\to {\mathrm{Br}}_{i}^{\prime }+V_{I}^{•}$	1.59 **−0.27**	${\mathrm{Br}}_{\mathrm{Br}}^{x}+h^{•}\to {\mathrm{Br}}_{i}^{x}+V_{I}^{•}$ ${\mathrm{Br}}_{\mathrm{Br}}^{x}+2h^{•}\to {\mathrm{Br}}_{i}^{•}+V_{I}^{•}$	**−0.12** **−0.10**
α-CsPbI_2_Br C1	$I_{I}^{x}\to I_{i}^{x}+V_{I}^{x}$ $I_{I}^{x}\to I_{i}^{\prime }+V_{I}^{•}$	1.73 **−0.26**	$I_{I}^{x}+h^{•}\to I_{i}^{x}+V_{I}^{•}$ $I_{I}^{x}+2h^{•}\to I_{i}^{•}+V_{I}^{•}$	0.39 **0.07**
	${\mathrm{Br}}_{\mathrm{Br}}^{x}\to {\mathrm{Br}}_{i}^{x}+V_{\mathrm{Br}}^{x}$ ${\mathrm{Br}}_{\mathrm{Br}}^{x}\to {\mathrm{Br}}_{i}^{\prime }+V_{\mathrm{Br}}^{•}$	1.93 0.69	${\mathrm{Br}}_{\mathrm{Br}}^{x}+h^{•}\to {\mathrm{Br}}_{i}^{x}+V_{\mathrm{Br}}^{•}$ ${\mathrm{Br}}_{\mathrm{Br}}^{x}+2h^{•}\to {\mathrm{Br}}_{i}^{•}+V_{\mathrm{Br}}^{•}$	0.80 0.96
	$I_{I}^{x}\to I_{i}^{x}+V_{\mathrm{Br}}^{x}$ $I_{I}^{x}\to I_{i}^{\prime }+V_{\mathrm{Br}}^{•}$	2.29 0.51	$I_{I}^{x}+h^{•}\to I_{i}^{x}+V_{\mathrm{Br}}^{•}$ $I_{I}^{x}+2h^{•}\to I_{i}^{•}+V_{\mathrm{Br}}^{•}$	1.16 0.84
	${\mathrm{Br}}_{\mathrm{Br}}^{x}\to {\mathrm{Br}}_{i}^{x}+V_{I}^{x}$ ${\mathrm{Br}}_{\mathrm{Br}}^{x}\to {\mathrm{Br}}_{i}^{\prime }+V_{I}^{•}$	1.37 **−0.08**	${\mathrm{Br}}_{\mathrm{Br}}^{x}+h^{•}\to {\mathrm{Br}}_{i}^{x}+V_{I}^{•}$ ${\mathrm{Br}}_{\mathrm{Br}}^{x}+2h^{•}\to {\mathrm{Br}}_{i}^{•}+V_{I}^{•}$	**0.03** **0.19**
α -CsPbI_2_Br P4/mmm (rc)	$I_{I}^{x}\to I_{i}^{x}+V_{I}^{x}$ $I_{I}^{x}\to I_{i}^{\prime }+V_{I}^{•}$	0.45 **−2.06**	$I_{I}^{x}+h^{•}\to I_{i}^{x}+V_{I}^{•}$ $I_{I}^{x}+2h^{•}\to I_{i}^{•}+V_{I}^{•}$	**−1.24** **−1.76**
	${\mathrm{Br}}_{\mathrm{Br}}^{x}\to {\mathrm{Br}}_{i}^{x}+V_{\mathrm{Br}}^{x}$ ${\mathrm{Br}}_{\mathrm{Br}}^{x}\to {\mathrm{Br}}_{i}^{\prime }+V_{\mathrm{Br}}^{•}$	0.54 **−1.34**	${\mathrm{Br}}_{\mathrm{Br}}^{x}+h^{•}\to {\mathrm{Br}}_{i}^{x}+V_{\mathrm{Br}}^{•}$ ${\mathrm{Br}}_{\mathrm{Br}}^{x}+2h^{•}\to {\mathrm{Br}}_{i}^{•}+V_{\mathrm{Br}}^{•}$	**−1.19** **−1.03**
	$I_{I}^{x}\to I_{i}^{x}+V_{\mathrm{Br}}^{x}$ $I_{I}^{x}\to I_{i}^{\prime }+V_{\mathrm{Br}}^{•}$	1.22 **−1.34**	$I_{I}^{x}+h^{•}\to I_{i}^{x}+V_{\mathrm{Br}}^{•}$ $I_{I}^{x}+2h^{•}\to I_{i}^{•}+V_{\mathrm{Br}}^{•}$	**−0.51** **−1.03**
	${\mathrm{Br}}_{\mathrm{Br}}^{x}\to {\mathrm{Br}}_{i}^{x}+V_{I}^{x}$ ${\mathrm{Br}}_{\mathrm{Br}}^{x}\to {\mathrm{Br}}_{i}^{\prime }+V_{I}^{•}$	**−0.23** **−2.07**	${\mathrm{Br}}_{\mathrm{Br}}^{x}+h^{•}\to {\mathrm{Br}}_{i}^{x}+V_{I}^{•}$ ${\mathrm{Br}}_{\mathrm{Br}}^{x}+2h^{•}\to {\mathrm{Br}}_{i}^{•}+V_{I}^{•}$	**−1.92** **−1.76**
β-CsPbI_2_Br Fmmm(Equatorial)	$I_{I}^{x}\to I_{i}^{x}+V_{I}^{x}$ $I_{I}^{x}\to I_{i}^{\prime }+V_{I}^{•}$	3.57 1.85	$I_{I}^{x}+h^{•}\to I_{i}^{x}+V_{I}^{•}$ $I_{I}^{x}+2h^{•}\to I_{i}^{•}+V_{I}^{•}$	1.73 0.75
	${\mathrm{Br}}_{\mathrm{Br}}^{x}\to {\mathrm{Br}}_{i}^{x}+V_{\mathrm{Br}}^{x}$ ${\mathrm{Br}}_{\mathrm{Br}}^{x}\to {\mathrm{Br}}_{i}^{\prime }+V_{\mathrm{Br}}^{•}$	3.40 1.51	${\mathrm{Br}}_{\mathrm{Br}}^{x}+h^{•}\to {\mathrm{Br}}_{i}^{x}+V_{\mathrm{Br}}^{•}$ ${\mathrm{Br}}_{\mathrm{Br}}^{x}+2h^{•}\to {\mathrm{Br}}_{i}^{•}+V_{\mathrm{Br}}^{•}$	1.58 1.68
	$I_{I}^{x}\to I_{i}^{x}+V_{\mathrm{Br}}^{x}$ $I_{I}^{x}\to I_{i}^{\prime }+V_{\mathrm{Br}}^{•}$	4.03 2.33	$I_{I}^{x}+h^{•}\to I_{i}^{x}+V_{\mathrm{Br}}^{•}$ $I_{I}^{x}+2h^{•}\to I_{i}^{•}+V_{\mathrm{Br}}^{•}$	2.21 1.22
	${\mathrm{Br}}_{\mathrm{Br}}^{x}\to {\mathrm{Br}}_{i}^{x}+V_{I}^{x}$ ${\mathrm{Br}}_{\mathrm{Br}}^{x}\to {\mathrm{Br}}_{i}^{\prime }+V_{I}^{•}$	2.94 1.04	${\mathrm{Br}}_{\mathrm{Br}}^{x}+h^{•}\to {\mathrm{Br}}_{i}^{x}+V_{I}^{•}$ ${\mathrm{Br}}_{\mathrm{Br}}^{x}+2h^{•}\to {\mathrm{Br}}_{i}^{•}+V_{I}^{•}$	1.11 1.21
β-CsPbI_2_Br I4/mcm(Apical)	$I_{I}^{x}\to I_{i}^{x}+V_{I}^{x}$ $I_{I}^{x}\to I_{i}^{\prime }+V_{I}^{•}$	1.25 **−1.37**	$I_{I}^{x}+h^{•}\to I_{i}^{x}+V_{I}^{•}$ $I_{I}^{x}+2h^{•}\to I_{i}^{•}+V_{I}^{•}$	**−1.29** **−1.16**
	${\mathrm{Br}}_{\mathrm{Br}}^{x}\to {\mathrm{Br}}_{i}^{x}+V_{\mathrm{Br}}^{x}$ ${\mathrm{Br}}_{\mathrm{Br}}^{x}\to {\mathrm{Br}}_{i}^{\prime }+V_{\mathrm{Br}}^{•}$	0.37 **−1.60**	${\mathrm{Br}}_{\mathrm{Br}}^{x}+h^{•}\to {\mathrm{Br}}_{i}^{x}+V_{\mathrm{Br}}^{•}$ ${\mathrm{Br}}_{\mathrm{Br}}^{x}+2h^{•}\to {\mathrm{Br}}_{i}^{•}+V_{\mathrm{Br}}^{•}$	**−1.43** **−1.34**
	$I_{I}^{x}\to I_{i}^{x}+V_{\mathrm{Br}}^{x}$ $I_{I}^{x}\to I_{i}^{\prime }+V_{\mathrm{Br}}^{•}$	1.31 **−0.58**	$I_{I}^{x}+h^{•}\to I_{i}^{x}+V_{\mathrm{Br}}^{•}$ $I_{I}^{x}+2h^{•}\to I_{i}^{•}+V_{\mathrm{Br}}^{•}$	**−0.49** **−0.37**
	${\mathrm{Br}}_{\mathrm{Br}}^{x}\to {\mathrm{Br}}_{i}^{x}+V_{I}^{x}$ ${\mathrm{Br}}_{\mathrm{Br}}^{x}\to {\mathrm{Br}}_{i}^{\prime }+V_{I}^{•}$	0.31 **−2.40**	${\mathrm{Br}}_{\mathrm{Br}}^{x}+h^{•}\to {\mathrm{Br}}_{i}^{x}+V_{I}^{•}$ ${\mathrm{Br}}_{\mathrm{Br}}^{x}+2h^{•}\to {\mathrm{Br}}_{i}^{•}+V_{I}^{•}$	**−2.22** **−2.13**

## Data Availability

No new data were created or analyzed in this study. Data sharing is not applicable to this article.

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
