# Peer review of "Redox Chemistry of the Subphases of α-CsPbI2Br and β-CsPbI2Br: Theory Reveals New Potential for Photostability"

_nanomaterials, 2023, doi:10.3390/nano13020276_

Round 1

Reviewer 1 Report

The manuscript is presenting an interesting Photostability research topic using first-principles computations. A few comments and questions are due before we proceed with this submission:

1-is there any proof for your assumption of α-CsPbI2Br P4/mmm, being the most stable pseudo-cubic phase? please note the abc values as not listed in table 1.

2- when calculating the deviation of α-CsPbI2Br and β-CsPbI2Br from Vegards Law,  the equatorial bonding is stronger which is new to me (Pb-I bond). would you correlate this to spin-orbit 242 coupling.

3-in introduction about the FA-based perovskite materials, a citation is essential to https://doi.org/10.1016/j.solmat.2022.112026

4-for any specific phase, is there any change between the oxidative capability of CsPbI2Br and CsPbI3?  what's the level of DOS for each? 

When the authors take my comments in and reply to my questions and revise the paper accordingly, i can reconsider my decision. do not skip any comment.

Author Response

We thank the reviewer for their attentive reading of our work and valuable insights which have helped us improve the work. Below we answer their queries point-by-point. We would like to point out that we have all worked on the grammar and typographical errors of the paper and have improved it in this regard also.

REVIEWER: 1-is there any proof for your assumption of α-CsPbI2Br P4/mmm, being the most stable pseudo-cubic phase? please note the abc values as not listed in table 1.

ANSWER: Within the limits of the accuracy of the theory we used P4/mmm is equal in stability to Inma; however, from the perspective of chemical logic it would seem like P4/mmm ought to be most stable since this is the only alpha phase-type where Pb-Br and Pb-I bonds would not strain one another. In all other phases some Pb-I bonds are aligned along the same axis as the Pb-Br bonds, the P4/mmm is the only case where this doesn’t happen: Pb-I are all aligned along a & b, Pb-Br are all aligned along c. Clearly, this will require further investigation; however, this doesn’t change the conclusions of our work. We clarified by modifying the statement to:

“Inma is equally stable to P4/mmm and is then energetically followed by Cm and Pmm2 (Figure 1). “

In the case of Table 1 we made changes to clarify since the a,b,c, was an accidental left over. Thus we reorganized Table 1 to be clearer. We left just lattice parameter a for alpha phases and parameter a and b for beta phases since these are most logical to include for experimental comparisons. We added this explanation to the table:

“Note that for α phases only the a lattice parameter is represented while for β phases the a and b parameters are shown.”

REVIEWER: 2- when calculating the deviation of α-CsPbI2Br and β-CsPbI2Br from Vegards Law,  the equatorial bonding is stronger which is new to me (Pb-I bond). would you correlate this to spin-orbit 242 coupling.

ANSWER: The equatorial bonding result is something that is not novel to our paper, we discovered it in https://link.springer.com/article/10.1007/s12039-020-01780-7  where the authors chose a apical and an equatorial alpha CsPbI2Br phase to study. In that paper the equatorial phase is 58 meV/f.u. more stable, our results agree with this. For pristine perovskites, SOC doesn’t seem to have a strong effect on the geometry, this was also noted in https://doi.org/10.1103/PhysRevMaterials.5.125408. We have included this paper in the draft as:

The limited effected of SOC on perovskite geometry has previously been observed in other works[46]

  1. Xue, H.; Brocks, G.; Tao, S. First-Principles Calculations of Defects in Metal Halide Perovskites: A Performance Comparison of Density Functionals. Phys. Rev. Mater. 2021, 5, 125408, doi:10.1103/PhysRevMaterials.5.125408.

REFVIEWER: 3-in introduction about the FA-based perovskite materials, a citation is essential to https://doi.org/10.1016/j.solmat.2022.112026

ANSWER: Thank you!! We cited it as citation [11] :

In the case of organic-inorganic perovskites (MAPbX3 and FAPbX3), this seems to be related to the counter cations which, while not being directly responsible for band gap behavior, screen carriers from being trapped or recombined [10], and thus the theoretical analysis of these materials is consequently more challenging, notably heterostructures, in particular, show promise in this regard[11].

  1. Hajjiah, A.; Gamal, M.; Kandas, I.; Gorji, N.E.; Shehata, N. DFT and AMPS-1D Simulation Analysis of All-Perovskite Solar Cells Based on CsPbI3/FAPbI3 Bilayer Structure. Sol. Energy Mater. Sol. Cells 2022, 248, 112026, doi:10.1016/j.solmat.2022.112026.

REVIEWER: 4-for any specific phase, is there any change between the oxidative capability of CsPbI2Br and CsPbI3?  what's the level of DOS for each?

ANSWER: We direct the attention of the reviewer to Table 3; however, we’d like to point out that we believe that the most important column here is the E_redox column since this gives us the hole (or electron) oxidation (or reduction) at the Fermi pinning of the free carrier.  We think this may be more objective as this difference isn’t notable when just looking at E_VBM and E_g by themselves. We note that the beta-equatorial species is significantly superior to all of the other mixed species.

To avoid confusion about this we added some clarification to the text:

“Surprisingly, the oxidative capability of CsPbI2Br differs very little from that of CsPbI3 in all phases except for β-CsPbI2Br Fmmm, which is more resilient than the other mixed phases. Adding an electrostatic correction did not essentially change the patterns observed.”

Reviewer 2 Report

Overall this is a solid manuscript with some interesting, novel findings. However, there were a few issues that should be addressed prior to publication.

l   A significant weakness of the study is the missing of future perspectives regarding the emerging new potential for photostability and/or other findings from your study.

l   Please add limitations section. It should report potential limitations of this specific study and the possibility of reproducibility in future setting.

l   There were many abbreviations in the manuscript. An addition of an abbreviation list may also be a good idea to make this clear.

Author Response

We thank the reviewer for their attentive reading of our work and valuable insights which have helped us improve the work. Below we answer their queries point-by-point. We would like to point out that we have all worked on the grammar and typographical errors of the paper and have improved it in this regard also.

REVIEWER:

 1. A significant weakness of the study is the missing of future perspectives regarding the emerging new potential for photostability and/or other findings from your study.

2. Please add limitations section. It should report potential limitations of this specific study and the possibility of reproducibility in future setting.

ANSWER: We thank the reviewer for these comments, given that we aren’t experimentalists we tried to rewrite our conclusion section to take these two aspects into account as much as is reasonably possible without speculating too much:

“In the present study, we found that the equatorial phase appears to be very stable as a mixed solid solution and is resilient against CFD formation whereas the apical phase is highly vulnerable to CFD formation and has a much higher mixing energy. This major difference in mixing energies implies that the material is likely capable of assuming the equatorial phase selectively, which means that selectively isolating the phase is a very real possibility in expanding the material scientist’s toolbox in engineering a stable perovskite photovoltaic cell. Given that the band gap seems to be somewhat wider for this phase than for the -  phase we have to conclude that the equatorial β-CsPbI2Br is ultimately a compromise of some photovoltaic performance in favor of photostability. An additional, potential difficulty is seen in the mechanism by which theory predicts the formation of such a phase: it originates from rotations of PbX6 octahedra in - , and thus a complex scheme might be necessary in successfully isolating this phase. Our calculations suggest that an experimental synthesis of the promising equatorial β-CsPbI2Br should probably bypass all -  phases to avoid random rotations into various β-CsPbI2Br phases.”

REVIEWER:  There were many abbreviations in the manuscript. An addition of an abbreviation list may also be a good idea to make this clear.

ANSWER: We are not certain if this idea is acceptable to the formatting of the journal; however, we placed such a list at the end of our work in case such a list can be included. We thank the reviewer for this idea.

Reviewer 3 Report

This manuscript assesses the destructive effects of holes on the mixed inorganic perovskite material. A wide variety of α-CsPbI2Br and β-CsPbI2Br phases, comparing their mixing enthalpies, exploring their oxidative properties, and calculating their hole-coupled and hole-free charge Frenkel defect formations by considering all possible channels of oxidation are also analyzed. The manuscript contains six keywords, four tables, seven figures plus nine supplementary figures, ten equations, and sixty-two references. Overall, it is a correct, complete, and well-conducted paper, although some remarks are made on different sections of the manuscript.

Keywords
The manuscript presents six keywords. For keywords, where possible, please use Medical Subject Headings terms (MeSH Terms). Strictly, only “density functional theory” is a MeSH term. Other MeSH terms proposed as keywords could be “calcium compounds”, “inorganic chemicals”, “oxides”, or “titanium”. However, these suggestions about MeSH terms as keywords are optional, not mandatory.

Other manuscript sections
To make text understanding easier, if the author's name appears in the text, place the reference number immediately after the name, not at the end of the sentence or paragraph. Why are authors cited in the text written in all capital letters, not only the first letter?

 References
Total number of the manuscript references: 62.
This is a complete and updated section. The reference format matches the journal’s reference format (ACS style guide).

Tables
Total number of the manuscript tables: 4.
The tables have appropriate titles and information.

Figures and supplementary figures
Total number of the manuscript figures: 7.
Total number of the manuscript supplementary figures: 9.
The figures have appropriate figure legends.

Equations
Total number of the manuscript figures: 10.
The equations have appropriate information.

Author Response

We thank the reviewer for their attentive reading of our work and valuable insights which have helped us improve the work. Below we answer their queries point-by-point. We would like to point out that we have all worked on the grammar and typographical errors of the paper and have improved it in this regard also.

REVIEWER:

Keywords
The manuscript presents six keywords. For keywords, where possible, please use Medical Subject Headings terms (MeSH Terms). Strictly, only “density functional theory” is a MeSH term. Other MeSH terms proposed as keywords could be “calcium compounds”, “inorganic chemicals”, “oxides”, or “titanium”. However, these suggestions about MeSH terms as keywords are optional, not mandatory.

ANSWER: MeSH doesn’t appear to have any terms relating to photovoltaics; however, we included “inorganic chemicals” in our list since this is appropriate.  Thanks for pointing this out!

REVIEWER: Other manuscript sections
To make text understanding easier, if the author's name appears in the text, place the reference number immediately after the name, not at the end of the sentence or paragraph. Why are authors cited in the text written in all capital letters, not only the first letter?

ANSWER: We fixed these issues as we encountered them. We believe we found them all.

Reviewer 4 Report

The present article entitled “Redox Chemistry of the subphases of α-CsPbI2Br and β-CsPbI2Br: Theory Reveals New Potential for Photostability”. In this study, α-CsPbI2Br and β-CsPbI2Br with novel Potential for Photostability theory. Thus, this reviewer recommends the publication of this work after addressing the following comments.

The abstract should provide some quantitative information.

The results and discussion sections should provide some important citations.

What about the conclusions section?

Typographical errors and superfluous spaces throughout the manuscript should be corrected.

Author Response

We thank the reviewer for their attentive reading of our work and valuable insights which have helped us improve the work. Below we answer their queries point-by-point. We would like to point out that we have all worked on the grammar and typographical errors of the paper and have improved it in this regard also.

REVIEWER: The abstract should provide some quantitative information.

ANSWER: We included quantitative information as:

“or to phase transform into equatorial β-CsPbI2Br which is resilient against the deleterious effects of hole-oxidation (energies of oxidation are >0 eV) and demixing (energy of mixing is <0 eV).”

In the abstract.

REVIEWER: The results and discussion sections should provide some important citations.

ANSWER: We added some new citations including:

“Previous calculations indicated that our PBE+D3 method is a reasonable compromise for calculating defect formation energies[46].”

“The limited effected of SOC on perovskite geometry has previously been observed in other works[46]”

  1. Xue, H.; Brocks, G.; Tao, S. First-Principles Calculations of Defects in Metal Halide Perovskites: A Performance Comparison of Density Functionals. Phys. Rev. Mater. 2021, 5, 125408, doi:10.1103/PhysRevMaterials.5.125408.

All of the values for CsPbI2Br somewhat underestimate the experimental value of 1.92 eV for CsPbI2Br[65–67].

  1. Mariotti, S.; Hutter, O.S.; Phillips, L.J.; Yates, P.J.; Kundu, B.; Durose, K. Stability and Performance of CsPbI2Br Thin Films and Solar Cell Devices. ACS Appl. Mater. Interfaces 2018, 10, 3750–3760, doi:10.1021/acsami.7b14039.
  2. Beal, R.E.; Slotcavage, D.J.; Leijtens, T.; Bowring, A.R.; Belisle, R.A.; Nguyen, W.H.; Burkhard, G.F.; Hoke, E.T.; McGehee, M.D. Cesium Lead Halide Perovskites with Improved Stability for Tandem Solar Cells. J. Phys. Chem. Lett. 2016, 7, 746–751, doi:10.1021/acs.jpclett.6b00002.
  3. Wang, Y.; Zhang, T.; Xu, F.; Li, Y.; Zhao, Y. A Facile Low Temperature Fabrication of High Performance CsPbI2Br All-Inorganic Perovskite Solar Cells. Sol. RRL 2018, 2, 1700180, doi:10.1002/SOLR.201700180.

The engineering of such a highly stable phase might be combined with other strategies such as non-stoichiometry [68] or the release of microstrain[69] to achieve a superior perov-skite-based photovoltaic material.

  1. Frolova, L.A.; Chang, Q.; Luchkin, S.Y.; Zhao, D.; Akbulatov, A.F.; Dremova, N.N.; Ivanov, A. V.; Chia, E.E.M.; Stevenson, K.J.; Troshin, P.A. Efficient and Stable All-Inorganic Perovskite Solar Cells Based on Nonstoichiometric CsxPbI2Brx (x > 1) Alloys. J. Mater. Chem. C 2019, 7, 5314–5323, doi:10.1039/C8TC04488K.
  2. Zheng, K.; Ge, J.; Liu, | Chang; Lou, Q.; Chen, | Xia; Meng, Y.; Yin, | Xu; Shixiao Bu, |; Liu, C.; Ziyi Ge, |; et al. Improved Phase Stability of CsPbI2Br Perovskite by Released Microstrain toward Highly Efficient and Stable Solar Cells. InfoMat 2021, 3, 1431–1444, doi:10.1002/INF2.12246.

REVIEWER: What about the conclusions section?

ANSWER: We previously made an error and mislabeled this section as “Discussion”. This has been fixed.

REVIEWER: Typographical errors and superfluous spaces throughout the manuscript should be corrected.

ANSWER: We have gone through the paper and worked on these issues as a group; many such corrections were made. We hope its to the Reviewer's satisfaction.